statistics/biomathematics/mathematical modelling

redaction, bias, biomedical, replication, statistics, hypothesis testing

**Author for correspondence:**
David Robert Grimes
e-mail: davidrobert.grimes@dcu.ie

# The new normal? Redaction bias in biomedical science

David Robert Grimes[1,2] and James Heathers[3]

[1]School of Physical Sciences, Dublin City University, Glasnevin, Dublin 9, Ireland
[2]Department of Oncology, University of Oxford, Oxford, Oxfordshire OX3 7DQ, UK
[3]Cipher Skin, Denver, CO, USA

  DRG, 0000-0003-3140-3278

A concerning amount of biomedical research is not reproducible. Unreliable results impede empirical progress in medical science, ultimately putting patients at risk. Many proximal causes of this irreproducibility have been identified, a major one being inappropriate statistical methods and analytical choices by investigators. Within this, we formally quantify the impact of inappropriate redaction beyond a threshold value in biomedical science. This is effectively truncation of a dataset by removing extreme data points, and we elucidate its potential to accidentally or deliberately engineer a spurious result in significance testing. We demonstrate that the removal of a surprisingly small number of data points can be used to dramatically alter a result. It is unknown how often redaction bias occurs in the broader literature, but given the risk of distortion to the literature involved, we suggest that it must be studiously avoided, and mitigated with approaches to counteract any potential malign effects to the research quality of medical science.

## 1. Introduction

Psychology was perhaps the first discipline to report a 'replication crisis', but there is increasing evidence that biomedical science is facing a similar problem of an even greater magnitude [1–3]. In a sample of medical studies performed between 1977 and 1990, flaws were evident in 20% of medical studies [4]. Another investigation of highly cited medical studies published between 1990 and 2003 found that while 45 originally claimed to find an effect, 16% were contradicted by further investigation, while another 16% reported effects stronger than subsequent studies allowed [5]. For landmark experiments in cancer research, the replication rate was an abysmal 11% [6].

Why might this situation be so prevalent in biomedical literature? Unedifying as it seems, fraud and poor practice explain part of the picture [7]. Inappropriate image manipulation was in 2006 estimated to occur in 1% of biological publications [8]—a figure likely to have grown as technology improves. A National

Institute of Health-funded study of early and mid-career scientists ($n = 3247$) found 0.3% admitted to falsification of data in the prior year, 6% failing to present conflicting evidence, and 15.5% admitted to changing study design, methodology or results following pressure from funders [9]. One recent overview suggested that 1–3% of scientists commit fraud, while questionable research practices occur in as much as 75% of published science [7]. As many of these figures require dishonest actors to honestly report scientific misconduct, they are almost certainly underestimates of true prevalence.

The reasons offered for the above are many, and are sometimes understood in terms of culpability—with 'unwitting error' on one end, 'the wholesale fabrication of data' on the other, and various questionable research practices and various methods of falsification somewhere in between. Discussions on these issues are many [10] but in our opinion a crucial source of error is deserving of increased awareness: dubious conclusions due to selective redaction of data included in experimental observations.

## 1.1. Redaction bias: a dangerous undertaking

Imagine a researcher taking individual measurements of $M$ ($m_1$, $m_2$, $m_3$, etc.), a hypothetical normally distributed quantity with a true mean of $A$ and a natural variability with a true standard deviation of $B$. If measurement error $e$ is assumed to be negligible, she observes the majority of her observations fall between $[A - 2B + e \quad A + 2B + e]$. But, on the fifth measurement, her measurement apparatus returns a value of $A - 6B$. Of course, the first step would be to check if this aberrant value is inflated because of a clearly identifiable mechanistic factor—perhaps a hardware, software, or calculatory error. But if one cannot be located, researchers often rely on heuristic rather than objective rules.

This sudden gatecrashing of the boundaries of expectation may require a decision about this data point's acceptability to be made instantaneously, perhaps while a measurement device with a long set-up time is still running, or the decision may be an unhurried one made during later analysis. Our researcher may have strong prior beliefs, or some, or none. She may have a great insight into the relevant problem space, or be encountering it for the first time. There may be sources of intrinsic heterogeneity within this measurement, and they may be known or unknown. She may work in a field which has a mechanistic definition of outlier values, and if such a definition exists it may be either well-justified or puzzlingly arbitrary. And, of course, the value returned for $m_5$ may be an unusual quantity measured accurately, revealing something startling or alarming or serendipitous about the phenomenon.

This situation is, of course, normal. *Every* experimentalist and research clinician encounters aberrant values, and becomes familiar with classifying data on a continuum of semi-formal or informal trustworthiness. This is particularly the case with human and animal data, and such classification is ubiquitous, necessary, often unobserved by other scientists. Attrition in clinical trials, particularly randomized controlled trials and longitudinal studies, has long been recognized as a serious issue in drawing inferences [11–16]. It is also frequently conducted in the absence of 'master' records. However, the process of redacting inaccurate data may cross over or even be replaced by the redaction of *unwanted* data. Redaction of data can be *systematic* due to some intrinsic fault in measurement or analysis—for example, a system which fails to register values beyond a certain threshold value. Alternatively, it can be due to accidental or deliberate *cherry-picking*, where only certain measurements are selected for inclusion in analysis. Finally, redaction can be an artefact of *attrition effect*—this we define as occurring when a specific subset of the experimental cohort has been removed relative to the control. For example, if only patients surviving a certain time-frame after an intervention are included in the sample and contrasted to a control without this stipulation. These types are illustrated in figure 1. In this work, we demonstrate that redacting results over or under an arbitrary threshold can be powerful and subtle enough in many cases to ostensibly support almost any claim in medicine and biomedical science.

## 1.2. The normal distribution and significance testing

In most biomedical fields, the implicit assumption is that biological variables approximately fit a normal (or related lognormal) distribution centred around a mean $\mu$ and standard deviation $\sigma$. The Gaussian distribution is exceptionally important, because even in situations where the underlying distribution is not normal, the central limit theorem states that the properly normalized sum of independent random variables tends towards a normal distribution, regardless of the underlying distribution of the original variables. Accordingly, with adequate sample size (typical $n > 30$), sample distributions can be presumed normal and analysed similarly.

Significance testing is perhaps the most widely used approach for hypothesis testing in biomedical science, and accordingly this is the focus of this work. In significance testing, one contrasts a sample

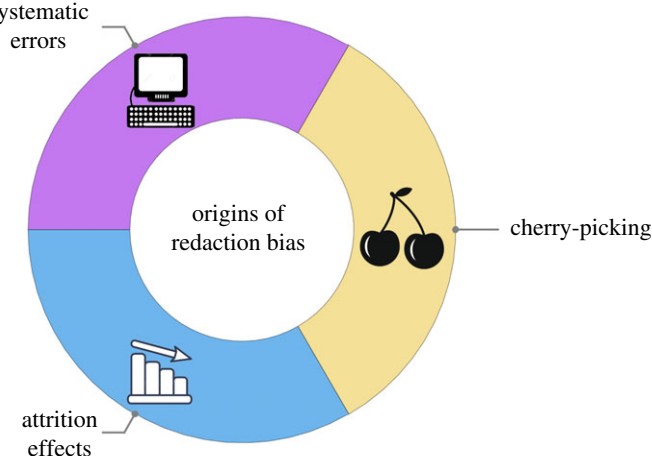

**Figure 1.** Origins of redaction bias.

mean against a known reference mean. In this process, a test statistic is determined from the sample distribution, and from this a $p$-value obtained. The $p$-value is defined as the probability, under the null hypothesis, that a test statistic at least as extreme as that would be observed. This is in reality an arbitrary number, but as a rule of thumb, Fisher suggested $p < 0.05$ merited deeper investigation. Parametric tests such as Student's $t$-test are frequently employed to ascertain whether a sample mean $\overline{X}$ with sample standard deviation $\sigma_s$ from sample size $n$ is significantly different from a known population with mean $\mu$. In the simplest case, the one sample $t$-test metric is

$$t = \frac{\overline{X} - \mu}{\sigma_s / \sqrt{n}}. \tag{1.1}$$

The significance level can be calculated from Student's $t$ distribution, for sample size $n$ with $n - 1$ degrees of freedom. Inferences drawn from naive significance testing, however, are fraught with pitfalls. Significance levels are arbitrary, and the misguided interpretation that $p < 0.05$ is a proxy for proof has been widely criticized [17]. Many experimenters still wrongly believe that the $p$-value is the probability that experimental results are due to chance, but this is not the case. Simply warning against this misinterpretation, however, has been deemed an 'abysmal failure' [18]. This is a problem likely compounded by the ease of modern statistics packages, which can readily run any test the user dictates, whether or not these are appropriate. Such mistaken understandings have led to the phenomena of $p$-hacking, where inappropriate manipulations are deployed to render results statistically significant [19–22], a practice that continues unabated in biomedical science.

Significant results do not quantify how impactful an intervention might be, nor does it reveal anything about clinical relevance. Abuses of this metric have led several journals to insist that investigators report other metrics such as effect size to quantify whether a statistically significant finding is clinically relevant. There are many instances where a highly significant result may have an effect size that renders it clinically negligible—for example, if $n$ is sufficiently high. In the era of large databases of data (such as genomic information), statistically significant findings can be yielded with no practical impact on clinical practice [23]. Effect size quantifies the strength of an apparent association, and there are several related definitions based on mean differences; for example, Glass's $\Delta$, defined as

$$\Delta = \frac{\mu_n - \mu}{\sigma_n}, \tag{1.2}$$

where $\mu_n$ is the sample mean, $\mu$ is the reference mean and $\sigma_n$ is the standard deviation of the sample. Is it worth noting that not only does $\Delta$ have no $n$ dependence, $\partial\Delta/\partial\mu = \partial\Delta/\partial\sigma = 0$, and so $\Delta$ is not dependent on standard deviation or mean either.

## 2. Methods

Consider a series of results which form an approximately normal distribution with mean $\mu$. We then consider a scenario where, due to redaction of results below a certain threshold, a portion of

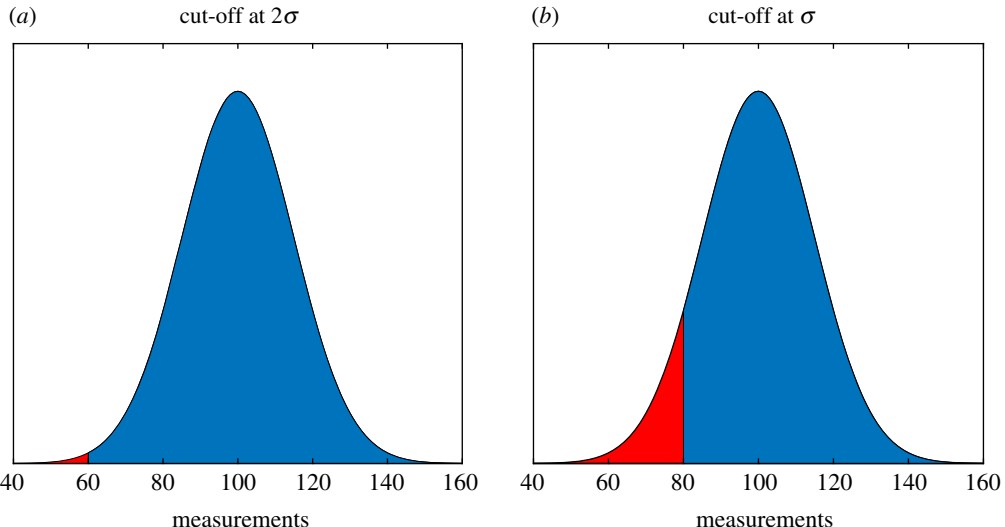

**Figure 2.** The effects of systematically jettisoning data below a given threshold to produce distorted Gaussian distributions. For a true distribution with $\mu = 100$ and $\sigma = 20$, the area in red is the extent of data jettisoned for (a) $2\sigma$ cut-off, corresponding to lower threshold of 60 for the data and (b) (a) $1\sigma$ cut-off, corresponding to lower threshold of 80 for the data. Impacts of this selection bias are discussed in the text.

experimental findings are disregarded from analysis. Truncated normal distributions and outlier removal have been considered by statisticians previously [24–26] and in this work, we explicitly derive a general identity for truncation as could be applied to biological datasets, where measurements below a given threshold are either deliberately or inadvertently disregarded from analysis, to ascertain how much impact this practice could have on biomedical results. We define this threshold in terms of the mean and standard deviation as $\mu - \omega\sigma$, where $\omega$ is a positive or negative constant, examples of which are given in figure 2. The sample mean in this distorted Gaussian can be shown to be given by

$$\mu_n = \begin{cases} \mu + \dfrac{\exp\left(-(\omega^2/2)\right)\sqrt{2/\pi}\sigma}{1 + \operatorname{erf}\left(\omega/\sqrt{2}\right)} & \text{if } \omega \geq 0 \\[2ex] \mu - \dfrac{\exp\left(-(\omega^2/2)\right)\sqrt{2/\pi}\sigma}{1 + \operatorname{erf}\left(-\omega/\sqrt{2}\right)} & \text{if } \omega < 0, \end{cases} \tag{2.1}$$

where erf is the error function. Where $\operatorname{erf}^{-1}$ is the inverse error function, it can also be shown that the displaced median after redaction is given by

$$m = \mu + \sqrt{2}\sigma\, \operatorname{erf}^{-1}\left(\frac{\operatorname{erfc}(\omega/\sqrt{2})}{2}\right). \tag{2.2}$$

Full derivations for these identities, and an explicit expression for the redacted standard deviation, are given in the electronic supplementary material, mathematical appendix. These parameters also can be applied to lognormal distributions and survival analysis, as the lognormal is intimately related to the standard Gaussian distribution where full mathematical details are given in the appendix. It is possible with this model to quantify how such redactions would impact conclusions drawn from research. We do this by simulating redaction impacts to both realistic biomedical and medical problems. Expanded simulations of redaction impacts on patient groups, including effect size, are also given in the electronic supplementary material. The identities here yield the values for a redacted mean and median, given perfect knowledge of $\mu$ and $\sigma$. When this is estimated from sample size $n$, the impacts of redaction can be even greater, as elucidated in illustrated examples here.

## 2.1. Biological and medical examples

To showcase how selection bias and distorted distributions might impede understanding of medical science, medically relevant examples typically encountered in cancer science were generated and the impact of selection bias quantified. These were, specifically,

1. ***In vitro*—Oxygen consumption:** Oxygen is a potent radio-sensitizer, and drugs that can reduce this and increase oxygen concentration in cancer are highly useful [27]. Consider extracellular flux analysis of plated cells with $s_H = 250 \pm 100$ pM of oxygen per cell per minute for untreated cells, with 25 plate repeats for a candidate drug. Here, we simulate the potential impact of redacting even one apparent outlier on conclusions drawn.

2. **Pre-clinical—Animal studies of therapeutic efficacy:** Murine experiments are typical in ascertaining the impacts of different agents on tumour growth. Tumour growth itself is highly variable. Here, we simulate a hypothetical experiment to examine whether cannabis-derived compounds might reduce tumour size, and simulate the impact on interpretation as individual mice are redacted from the analysis. Results from this simulated experiment are contrasted with a known control distribution where mean tumour diameter is $10 \pm 6$ mm in untreated mice.

3. **Human trial—Ostensible survival gain from ineffective intervention:** In this example, we consider a condition that follows a lognormal distribution with parameters $\mu = 6.2394$ and $\sigma = 1.0230$, corresponding to a median survival of 17.1 months (see mathematical appendix for details on lognormal conversion). If 300 patients are initially recruited for a small trial, but those who survive less than six months are excluded from analysis due to attrition effects, we can ascertain the impact of this on reported results.

For each example, normal (or related lognormal) distributions were generated in Matlab 2018 (Mathworks), centred on the mean with standard distribution. A normal (or related lognormal) distribution was accordingly fit to these illustrative examples. These datasets were then thresholded to remove points above or below $\omega$ standard distributions to simulate redaction, and a new normal (or lognormal) distribution was fit. A *t*-test was then performed, and the significance of the ostensible result calculated. Illustrations of spurious results are given here, and in the appendix, redactions are run 10 000 times to ascertain how often a false positive for significance was found for varying threshold values.

## 2.2. Effect sizes from redacted data

We can also calculate the likely effect size due to redaction, by applying equations (1.2) and (2.1) to differing degrees of redaction to ascertain likely impacts.

# 3. Results

## 3.1. In vitro: oxygen consumption

Figure 3 shows the data for 25 simulated plates. Before redaction, the sample mean and standard deviation of the entire sample are 230.13 μm and 85.05 μm, which for a two-tailed *t*-test yields $p = 0.25$. After redaction of the largest observation at $|\omega| = 2.38$ standard deviations away from the mean, the redacted mean of the 24 remaining observations is 219.40 μm with a standard deviation of 67.48 μm, and a two-tailed *t*-test in this instance $p = 0.036$. This sudden pivot to seeming significance after retraction of a single outlier might seem surprising, but it can also be inferred from equation (2.1); the actual cut-off is the nearest data value adjacent to the excluded point, which in this instance is the spheroid at 341.37 μm. This corresponds to an actual redaction threshold of $|\omega| = 0.91$, yielding a predicted value of $\mu_n = 217.93$ μm, in close agreement with the measured value. Effect size in this instance would be 0.45, a modest value.

### 3.1.1. Pre-clinical: animal studies of therapeutic efficacy

Figure 4 shows simulated experimental results of the new compound for 10 mice, sorted by tumour diameter at sacrifice. In this sample, $\mu = 9.6034$ mm and $\sigma = 7.48$ mm. When all 10 mice are considered, results are not significantly different from untreated controls with $10 \pm 6$ mm tumour diameter. Redacting the two greatest recorded diameters from the treatment group, however, yields a redacted sample mean of 6.37 mm with standard deviation 3.15 mm, which on a two-tailed *t*-test gives an illusion of high significant effects from the drug, and an apparently large effect size of 1.15. This corresponds to $|\omega| = 0.31$, and a predicted $\mu_n = 6.32$ mm, close to the measured value.

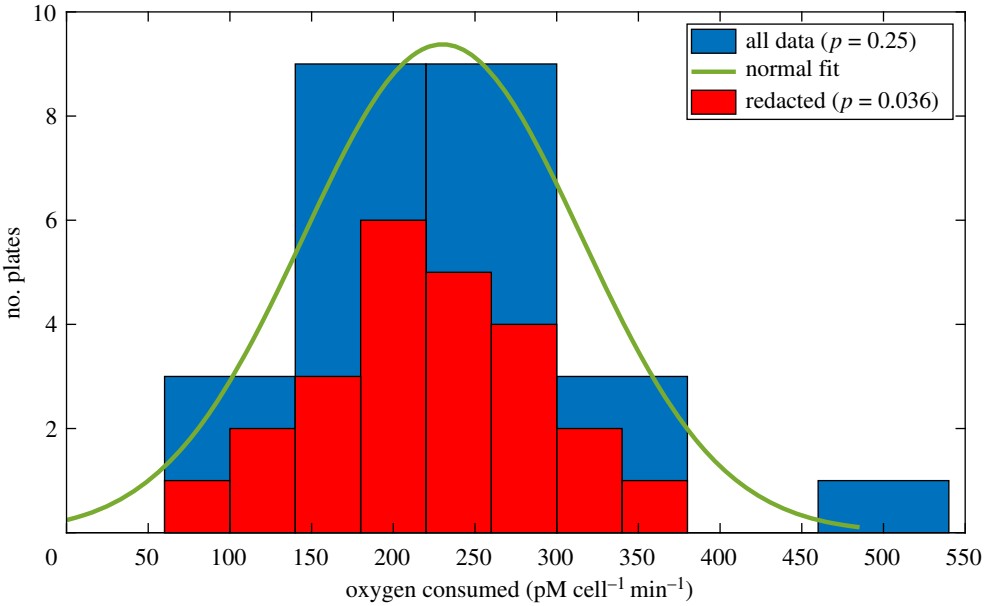

**Figure 3.** A non-effective drug being given to reduce oxygen consumption to plated cells. When all 25 repeats are considered, results are consistent with untreated value ($s_H = 250$ pM $O^2$ min$^{-1}$) but the redaction of the top-most value would cause an experimenter to wrongly reject the null value. Redacted histogram is shown with twice the number of bars for clarity.

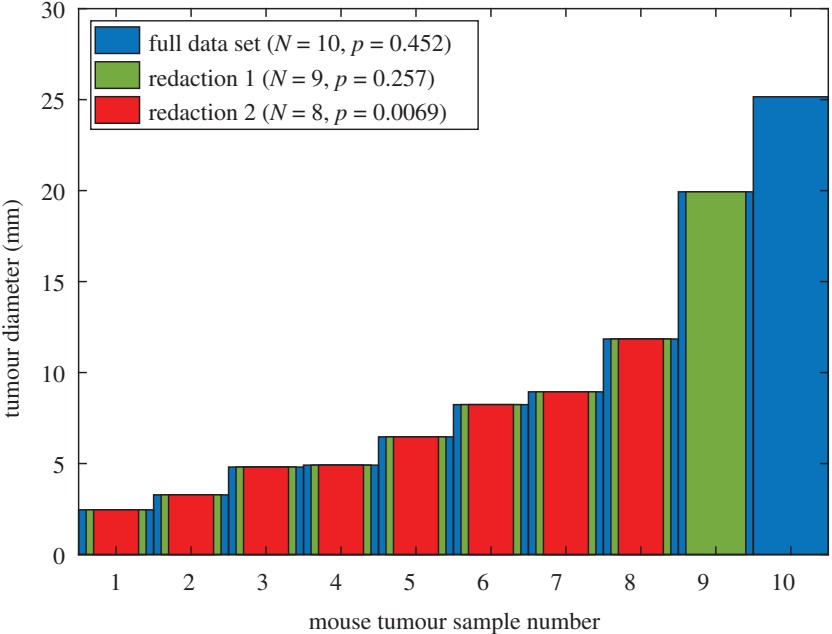

**Figure 4.** Pre-clinical results for an experimental drug on murine tumour size. Redacting the uppermost two mice from the analysis yields a significant result and large effect size. Individual mice are shown here, sorted by tumour diameter for clarity.

### 3.1.2. Human trial: ostensible survival gain from ineffective intervention

Figure 5 shows the Kaplan–Meier survival curves (depicting the fraction of surviving patients) for the entirety of the sample, and for a situation when patients not surviving beyond six months are excluded from the analysis. This corresponds to $\omega = 1$. For the all-patient cohort, median survival is $\exp(\mu)$ days, or 17.1 months. When those surviving under six months are excluded, equation (2.1) yields $\mu_n = 6.53$, corresponding to a median survival time of 22.9 months. A distribution fit to the simulated scenario in figure 5 yielded a lognormal with $\mu = 6.64$, in close agreement with theoretical prediction. This is statistically significantly different from $\mu$ ($p < 0.001$) with effect size 0.29. Redaction

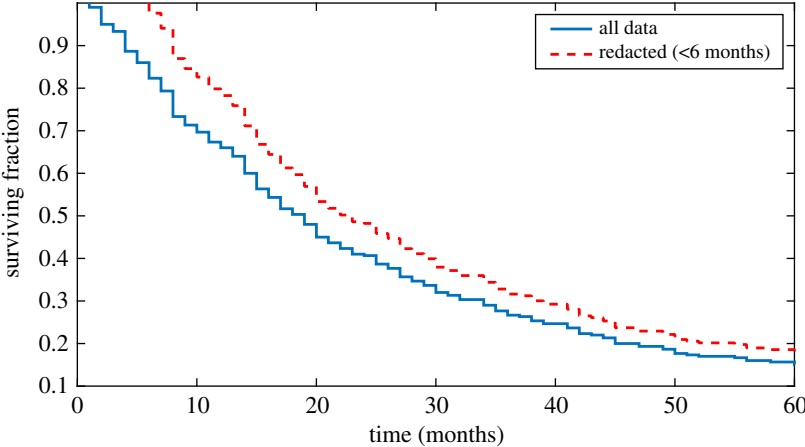

**Figure 5.** Kaplan–Meier survival curves for all patients ($N = 300$, blue solid line) and only patients surviving beyond six months ($N = 253$, red dashed line). The ostensible difference in survival is significant. See text for details.

of these patients would thus incorrectly lead an investigator to conclude that the intervention significantly increases survival time. It should be noted that such a redaction would be extremely poor practice, but inadvertent redactions could pivot on more subtle issues than survival time, such as exclusions due to a certain biomarker concentration or patient age.

## 3.2. Predicted impact of redaction on effect size

**Table 1.** Theoretical effect-size limits with varying degrees of redaction.

| cut-off ($\omega$) | theoretical $\Delta$ |
|---|---|
| ±0 | 0.798 |
| ±0.5 | 0.509 |
| ±1 | 0.288 |
| ±1.5 | 0.139 |
| ±2 | 0.055 |

## 4. Discussion

Whether intentional or inadvertent, redaction of data yields highly misleading results. In this paper, we have quantified how much different levels of data redaction will impact perceived results from normal and lognormal distributions, with the intention of illustrating how these missteps can be circumvented. Table 1 illustrates the minimum theoretical change in effect size with differing degrees of redaction. It is important to note that it is currently unknown how prevalent redaction itself is in biomedical literature, but it seems reasonable to presume that selective truncation of data leads to at least some of the problems with irreproducible research. There are of course instances when it might be appropriate to exclude data from analysis, but it is imperative that the reasons for the redaction are made clear, and that this excluded data is reported so that inappropriate censoring can be identified before dubious results take hold. The great physicist Richard Feynman once warned against the dangers of 'cargo-cult' science, that which apes the veneer of scientific investigation, advising that

> '... if you're doing an experiment, you should report everything that you think might make it invalid—not only what you think is right about it: other causes that could possibly explain your results; and things you thought of that you've eliminated by some other experiment, and how they worked—to make sure the other fellow can tell they have been eliminated. Details that could throw doubt on your interpretation must be given, if you know them. You must do the best you can—if you know anything at all wrong, or possibly wrong—to explain it. If you make a theory, for example, and advertise it, or put it out, then you must also put down all the facts that disagree with it, as well as those that agree with it.'

This is a fundamental principle of the scientific method that is too frequently ignored—the aim of investigation is not to prove ourselves right, but to present evidence in context, and for this reason we

must be wary of the misleading siren-song of redaction. Accidentally finding significance, as we see here, is too easily done—and engineering significance can be astonishingly straightforward with the removal of a few observations. This is only amplified by the problem of publication bias, where flimsy significant results are more readily published and garner more traction than reliable null results. This is of course itself perverse—it is every bit as important to know a drug does not work as to falsely believe it does, and yet only the latter is rewarded. This is something that must be urgently addressed if more trustworthy science and less wasted research efforts are the goal of scientific investigation.

The chief aim of this work is to explicitly demonstrate why great caution needs to be taken in biomedical science to avoid arriving at erroneous conclusions when using significance testing. While many of these problems have been long known and appreciated in statistical literature, they often are underappreciated in biomedical science. Specifically, the focus here is on inappropriate redacting of data above or below certain thresholds, and consequences of this. Awareness itself is critical, but there is much more that can be implemented to increase the reproducibility of published science. Specifying protocols for how the data will be analysed in advance of acquisition and ascertaining the statistical protocols to be used prior to this will tend to decrease selective redaction and dubious reporting [28].

But another part of the solution must be a cultural shift in biomedical science towards data-sharing, which when well-implemented galvanizes reproducibility [29]. There is still some reticence towards this; a 2018 survey found that less than 15% of researchers currently share either data or code, with data privacy the greatest cited concern [30]. Other research suggests that ineptitude with data curation is an issue for researchers [31], and suboptimal data curation can render shared data difficult to parse [32]. While there are moves towards greater data transparency and availability in several biomedical fields [33–36], an attitude in some fields that data sharing encourages 'research parasites' still endures [37] and needs to be redressed.

Pre-registration of protocols in clinical research too is crucial to maintain trust, especially as there can often be marked discrepancy between pre-registered approached and publications [38,39]. The involvement of statisticians prior to designing experiments and gathering data, and in the analysis of that data, would be extremely effective at circumventing many of the issues that arise in biomedical undertakings [40].

The structure of modern biomedical science often contributes to this—results are passed between groups and subgroups, reanalysed elsewhere, and fit to narrative by groups of co-authors who are dissociated from the experimental interface. An accurately measured, potentially informative outlier measured by a bench scientist may be passed to an analyst, and without the context of measurement, may instantly become a nuisance value, and potentially redacted. In general, even without redaction, small datasets are more easily swayed by outliers than larger collections, and for this reason more data is generally preferred. Equation (2.1) in this work yields the 'best-case' level of distortion from the true mean with redaction, but in practice distortion from the mean is even more extreme and pronounced with smaller samples.

Clearly, naive fixation on arbitrary statistical significance alone can dangerously mislead investigators, a fact that has been explicitly elucidated in recent years [17]. Significance testing in isolation can be misleading, largely because there is still widespread confusion over what significance actually means. Statistics arrived at must be seen in context; while large datasets are less prone to misleading bias than smaller ones, for example, false significance is more readily found in larger datasets, due to the inverse dependence of these metrics on sample size. This is only true for Fisherian approaches outlined in this work, and many of these pitfalls can be circumvented using different approaches to hypothesis testing, such as Bayesian analysis or likelihood ratios [18]. Effect sizes too, should inform interpretation of results derived. For ascertaining clinical impacts, absolute effect sizes are much more important than arbitrary $p$-values. While these are by no means new observations, the low levels of replicable research in biomedical science suggests lessons still must be learnt.

This leaves us with the serious question: in the entire scientific enterprise, how much data is being selectively redacted to engineer 'significant' results? This is a maddeningly hard question to answer, as it requires either direct observation of poor data handling practice (which is rarely published), or for researchers to report that this practice is taking place due to carelessness, confirmation bias, or dishonesty. There are clues, of course: for instance, one estimate [41] reveals that a tremendous amount of animals which are used in research (for mice, more than 75%) are never eventually reported in scientific outputs. While this total includes preliminary and early investigative work, experiments failed due to errors or unreliable laboratory procedures etc., the figure strongly suggests that a discrepancy between the amount of animals used in an experiment and the amount reported in an eventual publication is not subject to strong oversight. In other words, laboratory environments seem to be permissive of 'missing' data in other contexts.

In a psychological study [42], a retraction was provoked due to recoding of participant data between groups. This case is extremely unusual in so much as both sets of data were offered to external scientists investigating the veracity of the results, allowing them to clearly see the application of the issues discussed in this work. Considering the strong control over results offered by the simple redaction, it follows trivially that redacting data points from one group, and appending them to another to engineer an effect is a particularly powerful method of engineering false-positive results. Analysis using the fragility index has shown this to be an issue in the interpretation of a number of clinical trial results [43,44].

This work highlights and quantifies the problem of redaction, and why it is highly damaging to the undertaking of quality biomedical science. To prevent wasted research efforts and the chasing of spurious results, it is not enough to report summary statistics or redact data without clarity. Reasons for exclusion and inclusion must be reported, and ideally data should be made available for other researchers rather than jealously guarded. This is not a trivial problem to circumvent, and the impact goes far beyond dubious papers. As science is a collaborative effort, the elevation of false positives to scientific canon pushes scientists in wrong-headed directions, to our collective detriment. This injury is compounded by insult when one considers that sloppy research practices might actually garner rewards for inept researchers, at the expense of more diligent undertakings [3]. Worse again, poor research practices can even fuel misinformation [45,46] (and disinformation) around science and medicine, giving a veneer of respectability to wrong-headed positions. Pertinent examples of this range from the fraudulent research that deviously and wrongly linked the measles-mumps-rubella vaccine to autism [47,48], to the substandard trials that gave the false impression ivermectin was a viable COVID treatment [49,50]. The unsettling reality is that poor statistical practice renders swathes of biomedical research worse than useless. How we best address this is an open question, but a failure to do so threatens to fatally undermine scientific endeavour, to our collective detriment.

Data accessibility. The electronic supplementary material contains derivations of the identities in this work, and additional results and simulations. Data and relevant code for this research work are stored in Dryad, available at https://doi.org/10.5061/dryad.2v6wwpzp2 and at reviewer URL: https://datadryad.org/stash/share/Wx7Bs7lHZBnvdz1Xo_Zh40BzCrEnGpA2N4dOZVgJ3CM.

Authors' contributions. D.R.G. concept, derivations, simulations, writing and funding acquisition. J.H. concept, writing and editing.

Competing interests. We declare we have no competing interests.

Funding. D.R.G. is supported by the Wellcome Trust (grant no. 214461/A/18/Z). The authors would like to sincerely thank the reviewers for their insights and constructive criticisms which have largely improved this work.

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
