## [Peer Review File · Royal Society Open Science]

Review History

RSOS-210585.R0 (Original submission)

Review form: Reviewer 1

Is the manuscript scientifically sound in its present form?

No

Are the interpretations and conclusions justified by the results?

No

Is the language acceptable?

Yes

Do you have any ethical concerns with this paper?

No

Have you any concerns about statistical analyses in this paper?

Yes

Recommendation?

Reject

Comments to the Author(s)

In this submission, the authors discuss the notion of redaction bias as a source of bias in the biomedical literature, work out the statistical implications, and illustrate the problem with three examples. My major concern is that the statistical problem introduced here has been known for decades in the statistical literature as the one-tailed truncated normal distribution, but that the authors make no attempt to refer to the wider literature or to relate their work to what is already known. This also applies to work on outlier removal and robust statistics which is completely ignored but should be addressed and discussed.

Page 1

The abstract promises to present countermeasures, but these are not really discussed in the paper. They include: statistical protocols, use of robust analyses, standardization of measures, involvement of statisticians, data sharing, etc.

Page 2

The introduction raises important points, but lacks focus. It is unnecessarily long when it sets the stage (replication problems could be mentioned in one sentence) but then fails to cover relevant literature on the role of outlier removal, robust statistics, truncation models, pre-registration, attrition and reproducibility. I do recognize some of the origins of redaction bias, but they are not worked out very clearly. For instance, attrition effects are mentioned but the authors make no effort to relate this to the wider literature on missing data and attrition. Similarly, outlier removal has been often mentioned as a source of bias in the context of p-hacking but the writing here appears to suggest that the authors invented it.

Pages 3 and 4

The introduction of the PDF of the normal and the equation of the t test is not really needed and I would suggest the authors to replace it by an improved review of the existing literature.

Page 5.

The method is effectively a truncated normal and the results should be related to the wider literature. I also wondered whether anything is known the prevalence of the use of truncation.

Page 6.

It is rather obvious that the use of one-tailed truncation inflates the effect and hence the type 1 error rate. The example is rather limited however and given the strong conclusions I expected a much wider range of simulated (or formally computed) scenarios. Not that under a truncated normal, we could compute many statistics including the bias and the type 1 error rate.

Page 7

Note that excessive outliers (drawn from a mixture of normal and some skewed distribution) could also severely lower the power of normal based tests when studying genuine effects. This issue is not addressed.

Page 8.

The data generating processes underlying the plots were not clearly described.

Page 9

The discussion is strongly focused on the wider problem of failures to replicate but completely ignores earlier statistical work on outlier removal, truncated distributions, robust statistics, missing data, etc.

Page 10

There is no discussion of how common these practices actually are

Page 11

The authors discuss problems but no solutions

Review form: Reviewer 2 (David Colquhoun)

Is the manuscript scientifically sound in its present form?

Yes

Are the interpretations and conclusions justified by the results?

Yes

Is the language acceptable?

No

Do you have any ethical concerns with this paper?

No

Have you any concerns about statistical analyses in this paper?

Yes

Recommendation?

Major revision is needed (please make suggestions in comments)

Comments to the Author(s)

See attached file (Appendix A).

Decision letter (RSOS-210585.R0)

Dear Dr Grimes

The Editors assigned to your paper RSOS-210585 "The New Normal? Redaction bias in biomedical science" have made a decision based on their reading of the paper and any comments received from reviewers.

Regrettably, in view of the reports received, the manuscript has been rejected in its current form. However, a new manuscript may be submitted which takes into consideration these comments.

We invite you to respond to the comments supplied below and prepare a resubmission of your manuscript. Below the referees' and Editors' comments (where applicable) we provide additional requirements. We provide guidance below to help you prepare your revision.

Please note that resubmitting your manuscript does not guarantee eventual acceptance, and we do not generally allow multiple rounds of revision and resubmission, so we urge you to make every effort to fully address all of the comments at this stage. If deemed necessary by the Editors, your manuscript will be sent back to one or more of the original reviewers for assessment. If the original reviewers are not available, we may invite new reviewers.

Please resubmit your revised manuscript and required files (see below) no later than 20-Jan-2022. Note: the ScholarOne system will 'lock' if resubmission is attempted on or after this deadline. If

you do not think you will be able to meet this deadline, please contact the editorial office immediately.

Please note article processing charges apply to papers accepted for publication in Royal Society Open Science (<https://royalsocietypublishing.org/rsos/charges>). Charges will also apply to papers transferred to the journal from other Royal Society Publishing journals, as well as papers submitted as part of our collaboration with the Royal Society of Chemistry (<https://royalsocietypublishing.org/rsos/chemistry>). Fee waivers are available but must be requested when you submit your manuscript (<https://royalsocietypublishing.org/rsos/waivers>).

Thank you for submitting your manuscript to Royal Society Open Science and we look forward to receiving your resubmission. If you have any questions at all, please do not hesitate to get in touch.

on behalf of Dr John Ioannidis (Associate Editor) and Mark Chaplain (Subject Editor)
openscience@royalsociety.org

Associate Editor Comments to Author (Dr John Ioannidis):

Comments to the Author:

Thank you for this submission. As you will see from the reviewers' commentary, there are a number of areas that require attention before the paper may be considered ready for publication. Please make sure you thoroughly revise the paper before resubmitting as further revisions may not be possible.

Reviewer comments to Author:

Reviewer: 1

Comments to the Author(s)

In this submission, the authors discuss the notion of redaction bias as a source of bias in the biomedical literature, work out the statistical implications, and illustrate the problem with three examples. My major concern is that the statistical problem introduced here has been known for decades in the statistical literature as the one-tailed truncated normal distribution, but that the authors make no attempt to refer to the wider literature or to relate their work to what is already known. This also applies to work on outlier removal and robust statistics which is completely ignored but should be addressed and discussed.

Page 1

The abstract promises to present countermeasures, but these are not really discussed in the paper. They include: statistical protocols, use of robust analyses, standardization of measures, involvement of statisticians, data sharing, etc.

Page 2

The introduction raises important points, but lacks focus. It is unnecessarily long when it sets the stage (replication problems could be mentioned in one sentence) but then fails to cover relevant literature on the role of outlier removal, robust statistics, truncation models, pre-registration, attrition and reproducibility. I do recognize some of the origins of redaction bias, but they are not worked out very clearly. For instance, attrition effects are mentioned but the authors make no effort to relate this to the wider literature on missing data and attrition. Similarly, outlier removal

has been often mentioned as a source of bias in the context of p-hacking but the writing here appears to suggest that the authors invented it.

Pages 3 and 4

The introduction of the PDF of the normal and the equation of the t test is not really needed and I would suggest the authors to replace it by an improved review of the existing literature.

Page 5.

The method is effectively a truncated normal and the results should be related to the wider literature. I also wondered whether anything is known the prevalence of the use of truncation.

Page 6.

It is rather obvious that the use of one-tailed truncation inflates the effect and hence the type 1 error rate. The example is rather limited however and given the strong conclusions I expected a much wider range of simulated (or formally computed) scenarios. Not that under a truncated normal, we could compute many statistics including the bias and the type 1 error rate.

Page 7

Note that excessive outliers (drawn from a mixture of normal and some skewed distribution) could also severely lower the power of normal based tests when studying genuine effects. This issue is not addressed.

Page 8.

The data generating processes underlying the plots were not clearly described.

Page 9

The discussion is strongly focused on the wider problem of failures to replicate but completely ignores earlier statistical work on outlier removal, truncated distributions, robust statistics, missing data, etc.

Page 10

There is no discussion of how common these practices actually are

Page 11

The authors discuss problems but no solutions

Reviewer: 2

Comments to the Author(s)

See attached file

===PREPARING YOUR MANUSCRIPT===

While not essential, it will speed up the preparation of your manuscript proof if accepted if you format your references/bibliography in Vancouver style (please see

<https://royalsociety.org/journals/authors/author-guidelines/#formatting>). You should include DOIs for as many of the references as possible.

===PREPARING YOUR REVISION IN SCHOLARONE===

Author's Response to Decision Letter for (RSOS-210585.R0)

See Appendix B.

RSOS-211308.R0

Review form: Reviewer 1

Is the manuscript scientifically sound in its present form?

Yes

Are the interpretations and conclusions justified by the results?

Yes

Is the language acceptable?

Yes

Do you have any ethical concerns with this paper?

No

Have you any concerns about statistical analyses in this paper?

No

Recommendation?

Accept with minor revision (please list in comments)

Comments to the Author(s)

The revision dealt well with many issues raised by myself and the other reviewer, but the current version remains insufficient in how it refers to and discusses the wider literature on truncation models, outlier removal, and robust statistics. For instance, it fails to discuss Barnett & Lewis (1978, 1994) which was cited over 7000 times in its different editions (in Google Scholar and many in the biomedical literature) or similar work that has been deeply influential but not referred to in the paper. It also ignores the work on robust statistics (e.g., Huber, 2004; cited over 20000 times!)

and other important and influential literatures on how to deal properly with non-normality in data. Much of the current problems have been addressed dozens of times, and this needs to be acknowledged and discussed. Surely, it does not hurt to reinstate the important message that biases emerge from the very practices that the authors address but it is unacceptable to pretend the current readers all are sitting on a tiny island of the literature where they somehow missed the relevant literatures that thousands of others have cited and multiple dozens of others have actually already investigated to great lengths.

Review form: Reviewer 2 (David Colquhoun)

Is the manuscript scientifically sound in its present form?

Yes

Are the interpretations and conclusions justified by the results?

Yes

Is the language acceptable?

Yes

Do you have any ethical concerns with this paper?

No

Have you any concerns about statistical analyses in this paper?

No

Recommendation?

Accept with minor revision (please list in comments)

Comments to the Author(s)

I need to see the appendix. The link to Dryad returns "DOI not found".

Apart from this, I haven't many comments. The revisions are thorough and have improved the presentation greatly.

1. The use of the term 'atrophied' seems inaccurate to me. I suggest that it should be replaced.
2. The revision at lines 300-312 is a great improvement, I think. But surely it is not sensible to say "strictly speaking". It could be made even more clear if it were pointed out that the difference between the effect of sample size is dependent on whether you calculate the likelihood ratio by the p-equals method or the p-less-than method. Bayesian use the former but the distinction doesn't depend on whether you are Bayesian or not. It depends on whether you think that the analysis should reflect the evidence provided by the experiment you're analysing (p-equals) or long-term behaviour (p-less-than).

Decision letter (RSOS-211308.R0)

Dear Dr Grimes

On behalf of the Editors, we are pleased to inform you that your Manuscript RSOS-211308 "The New Normal? Redaction bias in biomedical science" has been accepted for publication in Royal Society Open Science subject to minor revision in accordance with the referees' reports. Please find the referees' comments along with any feedback from the Editors below my signature.

Please submit your revised manuscript and required files (see below) no later than 7 days from today's (ie 28-Oct-2021) date. Note: the ScholarOne system will 'lock' if submission of the revision is attempted 7 or more days after the deadline. If you do not think you will be able to meet this deadline please contact the editorial office immediately.

on behalf of Dr John Ioannidis (Associate Editor) and Mark Chaplain (Subject Editor)
openscience@royalsociety.org

Reviewer comments to Author:

Reviewer: 2

Comments to the Author(s)

I need to see the appendix. The link to Dryad returns "DOI not found".

Apart from this, I haven't many comments. The revisions are thorough and have improved the presentation greatly.

1. The use of the term 'atrophied' seems inaccurate to me. I suggest that it should be replaced.
2. The revision at lines 300-312 is a great improvement, I think. But surely it is not sensible to say "strictly speaking". It could be made even more clear if it were pointed out that the difference between the effect of sample size is dependent on whether you calculate the likelihood ratio by the p-equals method or the p-less-than method. Bayesian use the former but the distinction doesn't depend on whether you are Bayesian or not. It depends on whether you think that the analysis should reflect the evidence provided by the experiment you're analysing (p-equals) or long-term behaviour (p-less-than).

Reviewer: 1

Comments to the Author(s)

The revision dealt well with many issues raised by myself and the other reviewer, but the current version remains insufficient in how it refers to and discusses the wider literature on truncation models, outlier removal, and robust statistics. For instance, it fails to discuss Barnett & Lewis (1978, 1994) which was cited over 7000 times in its different editions (in Google Scholar and many in the biomedical literature) or similar work that has been deeply influential but not referred to in the paper. It also ignores the work on robust statistics (e.g., Huber, 2004; cited over 20000 times!) and other important and influential literatures on how to deal properly with non-normality in data. Much of the current problems have been addressed dozens of times, and this needs to be acknowledged and discussed. Surely, it does not hurt to reinstate the important message that biases emerge from the very practices that the authors address but it is unacceptable to pretend the current readers all are sitting on a tiny island of the literature where they somehow missed the relevant literatures that thousands of others have cited and multiple dozens of other have actually already investigated to great lengths.

===PREPARING YOUR MANUSCRIPT===

one version should clearly identify all the changes that have been made (for instance, in coloured highlight, in bold text, or tracked changes);

===PREPARING YOUR REVISION IN SCHOLARONE===

To revise your manuscript, log into <https://mc.manuscriptcentral.com/rsos> and enter your Author Centre - this may be accessed by clicking on "Author" in the dark toolbar at the top of the

page (just below the journal name). You will find your manuscript listed under "Manuscripts with Decisions". Under "Actions", click on "Create a Revision".

-- If you are requesting an article processing charge waiver, you must select the relevant waiver option (if requesting a discretionary waiver, the form should have been uploaded, see 'File upload' above).

-- If you have uploaded any electronic supplementary (ESM) files, please ensure you follow the guidance at <https://royalsociety.org/journals/authors/author-guidelines/#supplementary-material> to include a suitable title and informative caption. An example of appropriate titling and captioning may be found at https://figshare.com/articles/Table_S2_from_Is_there_a_trade-off_between_peak_performance_and_performance_breadth_across_temperatures_for_aerobic_scope_in_teleost_fishes_/3843624.

At the 'Review & submit' step, you must view the PDF proof of the manuscript before you will be able to submit the revision. Note: if any parts of the electronic submission form have not been

completed, these will be noted by red message boxes - you will need to resolve these errors before you can submit the revision.

Author's Response to Decision Letter for (RSOS-211308.R0)

See Appendix C.

Decision letter (RSOS-211308.R1)

Dear Dr Grimes,

I am pleased to inform you that your manuscript entitled "The New Normal? Redaction bias in biomedical science" is now accepted for publication in Royal Society Open Science.

on behalf of Dr John Ioannidis (Associate Editor) and Mark Chaplain (Subject Editor)
openscience@royalsociety.org

Appendix A

Comments on RSOS-210585

It is useful to have a quantitative idea of the effects of omitting data points. This paper provides that. The science is fine, but the presentation of the results could be much clearer. I have made suggestions about how to do this.

1. Table 1 needs a much more informative title, and description in the text. It isn't clear exactly what it's telling the reader. The present title, "Variation of effect size with sample size" has no clear meaning for me. I'm puzzled that Table 1, and the rest of the paper, seem to ignore entirely the Jeffreys-Lindley phenomenon. The whole paper is based on "statistically-significant means $p < 0.05$ ". You don't need to be Bayesian to see that this doesn't make sense. If you calculate the likelihood ratio as a proper way to judge the relative merits of two rival hypotheses, e.g, $\mu = 0$ versus $\mu = \hat{\mu}$ then it's apparent that, if the likelihood ratio is calculated by the *p-equals* method, rather than by the *p-less-than* method, then the dependence of the likelihood ratio on sample size is quite different. For example, in Fig 2 of Colquhoun 2019, $1/(1+L)$ is plotted against sample size and if this quantity is interpreted as a false positive risk (for prior odds = 1) it's seen that for large enough sample sizes (large enough power) the evidence, for any fixed p value, always tends to favour the null. This makes much more sense than the frequentist approach because when power is very high all p values would be very small, so observing a p value as *big* as 0.05 would favour the null hypothesis.

I'm not suggesting that the paper should be rewritten using likelihood ratios, or false positive risk, as the criterion for judging whether or not an observed difference is or isn't a result of sampling error. But more than enough has been written on the limitations of the p value approach that it would be helpful to them reader to include a discussion of the Jeffreys-Lindley effect so that the assumptions made in the paper are clearer to readers.

2. Equation 3 Needs more explanation, and a reference. The quantities are not defined at the point eq 3 appears: μ_n is not defined until further down page 5, and σ_n is not defined until you get to the appendix. It might also be worth mentioning that effect sizes that are normalised by σ are not of any clinical relevance.

3. In figure 2, the value of ω is not clear at all. Shouldn't " $\mu - 2\omega\sigma$ " in the legend be written as $\mu - 2\sigma$ *i.e.* the value for $\omega = 2$?

4. page 6, line 42. And page 12, line 16. "Erf is the standard error function". The use of "standard error" here is potentially confusing. Wouldn't it be better to say something like "here erf() is the error function, defined as . . .". Also erf() is normally spelled with lower case, like exp().

5. page 6, line 51. " μ_n is the limit of the warped mean as sample population, n , tends to infinity.". Wouldn't "censored" be a better word than "warped"? And wouldn't it be better to describe it as the true value, insofar as it assumes that μ and σ are known. There is no need to describe it as a limit (in fact it would only be a limit if the estimator were unbiased). The estimate using sample estimates of μ and σ will be

variable but it isn't obvious (to me) that it will be biased so I'm not sure that it's right to describe it as giving "minimum changes from the mean due to redaction".

6. page 7, line 38 " $< -2\omega$ " -again ω is not defined (see para 3, above). The example would be much clearer if were put as follows. Before redaction the sample mean and standard error are *** and ***, hence $t_{df} = \text{***}$ and $p = \text{***}$. After redaction of the largest observation (which is the only observation which is more than two 2 sample standard deviations above the sample mean) these values become ***. Presenting it this way would make it much clearer the extent to which the increase in t after redaction results from changes in sample standard deviation and to what extent it results from changes in its numerator.

7. Figure 4. Should the ordinate be *reduction* in tumour diameter, rather than tumour diameter? It's not at all clear where the control groups appear in the examples. Likewise at the top of page 10, it's not at all clear what "with $10 \pm 6\text{mm}$ tumour diameter" refers to.

8. Page 10, line 17. "theoretical $u_n = 6.53$ " should presumably to μ_n . A bit more explanation of what this should be compared with would make this clearer. A bit of explanation of the gaps in the red curve in Figure 5 would help the reader a lot. I wonder how realistic this example is: who on earth would exclude patients "not surviving beyond 6 months"? What reason could be given for doing something so apparently silly as that?

9. page 11, line 13. "Equation 4 in this work yields the 'best-case' in larger data sets, due to the inverse dependence of these metrics on sample size". The meaning of this isn't clear to me (see also para 5, above). Likewise "effect size is more often a better metric to assess impact of an intervention". I can't recall reading any paper in which the effect size is *not* stated, though sometimes it's given only in standardised form which makes judgement of clinical significance difficult.

10. page 11, table 2. More verbal explanation of what the table tells you is needed.

Line 31. "false significance is more readily found in larger data sets, due to the inverse dependence of these metrics on sample size". That's true only for the Fisherian approach, It isn't true for either the likelihood ratio approach or the Bayesian approach (see para 1, above).

Line 34. "effect size is more often a better metric to assess impact of an intervention". Here (and elsewhere) it should be made clear whether "effect size" refers to normalised effect size or absolute effect size. Only the latter is useful for judging clinical significance. The former is much like saying that the z score is a "better metric to assess impact of an intervention". I'm not sure how useful this comment is.

11. page 12, line 4. "redacting data points from one group, and appending them to other . . .". A reference to the fragility index might be appropriate here.

Appendix

12. I've tried to evaluate eqs 6 and 7 in R. I'd like to check against the authors' code, but it doesn't seem to be available.

13. page 14, line 45. Clearly eq. 9 is a measure of skewness (though not the conventional measure), but please explain why it is "a measure of how many patients are required in total for a trial to generate a spurious result"

14. page 15. I'll admit to having to struggle a bit to follow the results of the simulations. This is partly a result of the curious properties of ω as a measure of censorship - no censoring corresponds to $\omega \rightarrow \infty$. More importantly, it's partly a result of the fact that it isn't explained clearly exactly what is tested in the t tests. Is the sample from the uncensored distribution tested against the sample from the censored population in a two-sample t test?

Figure 6 is presumably calculated, not found via simulation, so it should not appear under the simulation heading. Please move it earlier, together with the first paragraph on page 15, and provide a verbal description of what it shows. Also please say exactly how the curves are calculated.

It's also worth mentioning again that the results hold only if you are happy to ignore the Jeffreys-Lindley effect and the likelihood ratio approach.

Figure 7. Again please provide a longer verbal description of what the Figure shows.

I wonder whether it's sensible to start the curves at $\omega = 0$. It's hard to imagine anyone redacting all the data below the mean.

Minor points

Italicisation of symbols is erratic throughout.

Page 3, line 4. Surely unexplored, not explored

Page 4, line 4 "normative circumstances". Does this mean "usually"?

Page 4, line 32 "expectation value"

Page 4, line 50-53 "statistically significant"

Page 5, line 18. Reword to avoid p -value as first word in a sentence.

Page 5, line 31-32.

Page 7, line 18. "itself highly variable" reads oddly. Suggest "tumour growth, which is highly variable"

Page 7, line 24 "ineffective", not "ineffectual".

Page 10, line 4 The reference to Figure 5 should presumably be to Figure 4.

Appendix B

Response to reviewers

Dr David Robert Grimes,
Davidrobert.grimes@dcu.ie
11th August 2021

Re: RSOS-210585

Dear Profs Sandwrs, Ioannidis, reviewers, and journal team,

Thank you so much for taking the time to evaluate this manuscript, and in particular to the reviewers for their extremely helpful comments. Both reviewers have given us impetus to rewrite and improve the work, and we would like to thank them both here and in the acknowledgments – if they are amenable to revealing their identity we would be very happy to name them in any final version, and for now they are credited as the reviewers.

Reviewer 1 gave us very useful feedback, and we were remiss in not clarifying precisely our ambitions with this work, nor reflecting the wider statistical literature. We have now modified it to clarify the scope, limitations, and need for this analysis in biomedical science. Reviewer 2 had extremely perceptive insights, and we have tried to incorporate all their suggestions. Please do let me know if there's anything further we can provide. Many heartfelt thanks again.

Yours,

David (on behalf of both authors)

Replies to reviewer 1

1. *"In this submission, the authors discuss the notion of redaction bias as a source of bias in the biomedical literature, work out the statistical implications, and illustrate the problem with three examples. My major concern is that the statistical problem introduced here has been known for decades in the statistical literature as the one-tailed truncated normal distribution, but that the authors make no attempt to refer to the wider literature or to relate their work to what is already known. This also applies to work on outlier removal and robust statistics which is completely ignored but should be addressed and discussed."*

We are grateful for the review's insight, and would of course be happy to include any references the reviewer thinks we ought to, and we would be grateful if the reviewer had specific suggestions on this aspect. Searching the term in statistics sources, we have added a reference to the truncated normal from Johnson, Kotz, and Balakrishnan's text "Continuous Univariate Distributions Volume 1", and alluded to the fact that a deeply related problem has been considered by statisticians before, amending our text to read:

"General forms for truncated normal distributions have been considered by statisticians previously [22] and in this work, we explicitly derive an identity for truncation as could be applied to biological data sets, where measurements below a given threshold are either deliberately or inadvertently disregarded from analysis, to ascertain how much impact this practice could have on biomedical results."

It is crucial to note that this paper has been written specifically for a biomedical audience, and while much of this is likely well known to statisticians, it is not sufficiently well known in medicine and biology. This can be demonstrated by searching the term the reviewer suggests on PubMed, which garners only 46 results. These are mainly from statistics journals, much of them dealing with fixed effect and bootstrapping models do not seem directly applicable to the concept we are focusing on in this work of data tampering, whether intentional or not. Outlier removal is fascinating, and we were able to delve into this at the reviewer's suggestion on pubmed, but most of the results we found were not directly applicable to this work. Again, we could be very grateful if the reviewer would care to give us specific suggestions on what ought to be considered. Some of these points are recurrent, and are addressed in more detail in subsequent sections of the reply.

One major issue we identified is that we did not distinguish from general redaction and the very specific truncated form we considered here (see also: point 3) – we have changed the text in several places to emphasise this, and the inherent limitations of our work, including at the end of the introduction:

“In this work, we demonstrate that redacting results over or under an arbitrary threshold can be powerful and subtle enough in many cases to ostensibly support almost any claim in medicine and biomedical science.”

- 2. “The abstract promises to present countermeasures, but these are not really discussed in the paper. They include: statistical protocols, use of robust analyses, standardization of measures, involvement of statisticians, data sharing, etc.”*

These are excellent points, and we were remiss for not elucidating these excellent suggestions – our chief goal was to outline for biomedical scientists the consequences of jettisoning data (see response to point 10 too) in the hope this could discourage the practice. But the reviewer is absolutely right that more rigorous ways much be suggested, and we have added the following text to the discussion:

“The chief aim of this work is to explicitly demonstrate why great caution needs to be taken in biomedical science to avoid arriving at erroneous conclusions. While many of these problems have been long known and appreciated in statistical literature, they often are underappreciated in biomedical science. Specifically, the focus here is on inappropriate redacting of data above and below certain thresholds, and consequences of this. Awareness itself is critical, but there is much more that can be implemented to increase the reproducibility of published science. Specifying protocols for how the data will be analysed in advance of acquisition and ascertaining the statistical protocols to be used prior to this will tend to decrease selective redaction and dubious reporting [22].

But another part of the solution must be a cultural shift in biomedical science towards data-sharing, which when well implemented galvanises reproducibility [23]. There is still some reticence towards this; a 2018 survey found that less than 15% of researchers currently share either data or code, with data privacy the greatest cited concern [24]. Other research suggests that ineptitude with data curation is an issue for researchers [25], and suboptimal data curation can render shared data difficult to parse [26]. While there are moves towards greater data transparency and availability in several biomedical fields [27–30], an attitude in some fields that data sharing encourages “research parasites” still endures [31] and needs to be redressed.

Preregistration of protocols in clinical research too is crucial to maintain trust, especially as there can often be marked discrepancy between pre-registered approaches and publications [32, 33]. The involvement of statisticians prior to designing experiments and gathering data, and in the analysis of that data, would be extremely effective at circumventing many of the issues that arise in biomedical undertakings [34]. “

- 3. “The introduction raises important points, but lacks focus. It is unnecessarily long when it sets the stage (replication problems could be mentioned in one sentence) but then fails to cover relevant literature on the role of outlier removal, robust statistics, truncation models, pre-registration, attrition and reproducibility. I do recognize some of the origins of redaction bias, but they are not worked out very clearly. For instance, attrition effects are mentioned but the authors make no effort to relate this to the wider literature on missing data and attrition. Similarly, outlier removal has been often mentioned as a source of bias in the context of p-hacking but the writing here appears to suggest that the authors invented it.”*

Apologies, this was never our intention – our goal was to make clear that this practice can skew things in biomedical research, for an audience who might not otherwise consider it. We have revised the introduction markedly to make it more readable and to clarify this is not a new problem, but one that needs to be very carefully considered in medical science. I think the problem in large part was our definition of “attrition” – we wrongly used this to refer to a specific type of attrition where the sample has lost a particular subset of its distribution relative to the control, rather than the more general redaction we referred to earlier. We were remiss in our phrasing – we have now clarified the introduction (see also point 4) and the specific changes relevant to this point as a clarification that data attrition is a long standing problem, and a qualification of what we mean by what we define as “attrition effects” for clarity:

“Attrition in clinical trials, particularly randomised controlled trials and longitudinal studies, has long been recognised as a serious issue in drawing inferences [11–16].”

*“Finally, redaction can be an artefact of **Attrition effect** - this we define as occurring when a specific subset has been atrophied between the control and sample. For example, if only patients surviving a certain time-frame after an intervention are included in the sample and contrasted to a control without this stipulation.”*

- 4. “The introduction of the PDF of the normal and the equation of the t test is not really needed and I would suggest the authors to replace it by an improved review of the existing literature.”*

We mainly agree with the reviewer. We are however acutely aware from experience that many colleagues do not know precisely why they are conducting a significance test, nor what it entails. We think therefore have relegated the pdf of the normal to supplementary, but kept a condensed version of the t-test and surrounding discussion in the text. Please also see response to review 2, point 2 also.

- 5. “The method is effectively a truncated normal and the results should be related to the wider literature. I also wondered whether anything is known the prevalence of the use of truncation.”*

Please see the response to point 1,2,3 and point 10.

6. *“It is rather obvious that the use of one-tailed truncation inflates the effect and hence the type 1 error rate. The example is rather limited however and given the strong conclusions I expected a much wider range of simulated (or formally computed) scenarios. Not that under a truncated normal, we could compute many statistics including the bias and the type 1 error rate.”*

Again, we don't disagree with the reviewer, as detailed in points 1 and 4. What we are concerned about however is a very specific application that occurs in biomedical science, where results below or above a certain threshold are jettisoned. The paper explicitly calculates the quantitative ramifications of this on biomedical literature, to quantify how misleading this can be. We have now clarified this in the introduction and discussion rewrites, and a much expanded literature presentation.

7. *“Note that excessive outliers (drawn from a mixture of normal and some skewed distribution) could also severely lower the power of normal based tests when studying genuine effects. This issue is not addressed.”*

This is true, but not the intention of this work, as mentioned in points 1 and 6. We have clarified this in this iteration of the manuscript as we had been much too vague initially.

8. *“The data generating processes underlying the plots were not clearly described.”*

Apologies, we should have specified this. The following text has been added to the method section:

“For each example, normal (or related log-normal) distributions were generated in MATLAB 2018 (Mathworks), centred on the mean with standard distribution. A normal (or related log normal) distribution was accordingly fit to these illustrative examples. These data sets were then thresholded to remove points above or below w standard distributions to simulate redaction, and a new normal (or log-normal) distribution was fit. A t-test was then performed, and the significance of the ostensible result calculated. Illustrations of spurious results are given here, and in the appendix, redactions are run 10,000 times to ascertain how often a false positive for significance was found for varying threshold values.”

9. *“The discussion is strongly focused on the wider problem of failures to replicate but completely ignores earlier statistical work on outlier removal, truncated distributions, robust statistics, missing data, etc.”*

We have addressed this somewhat in the new introduction and discussion (point 3), and specifically clarified with this. Please see also response to reviewer 2, point 11:

“While many of these problems have been long known and appreciated in statistical literature, they often are underappreciated in biomedical science.”

10. *“There is no discussion of how common these practices actually are.”*

The truthful answer is we don't know, and there is no reliable literature on it. We have added to the discussion...

“It is important to note that it is currently unknown how prevalent redaction itself is in biomedical literature, but it seems reasonable to presume that selective truncation of data leads to at least some of the problems with irreproducible research.”

11. *“The authors discuss problems but no solutions.”*

The chief intention of this work is to specifically illustrate the problem for a biomedical audience to illustrate why it results in dubious conclusions. However, we appreciate the reviewers point, and have expanded upon this – please see point 2 for details, and reviewer 2 points 2,11, and 13.

Replies to reviewer 2

1. *“It is useful to have a quantitative idea of the effects of omitting data points. This paper provides that. The science is fine, but the presentation of the results could be much clearer. I have made suggestions about how to do this.”*

We are extremely grateful to the review for their insights, and have responded to their points here.

2. *“Table 1 needs a much more informative title, and description in the text. It isn’t clear exactly what it’s telling the reader. The present title, “Variation of effect size with sample size“ has no clear meaning for me. I’m puzzled that Table 1, and the rest of the paper, seem to ignore entirely the Jeffreys-Lindley phenomenon. The whole paper is based on “statistically-significant means $p < 0.05$ ”. You don’t need to be Bayesian to see that this doesn’t make sense. If you calculate the likelihood ratio as a proper way to judge the relative merits of two rival hypotheses, e.g. $\mu = 0$ versus $\mu = \hat{\mu}$ then it’s apparent that, if the likelihood ratio is calculated by the p -equals method, rather than by the p -less-than method, then the dependence of the likelihood ratio on sample size is quite different. For example, in Fig 2 of Colquhoun 2019, $1/(1+L)$ is plotted against sample size and if this quantity is interpreted as a false positive risk (for prior odds = 1) it’s seen that for large enough sample sizes (large enough power) the evidence, for any fixed p value, always tends to favour the null. This makes much more sense than the frequentist approach because when power is very high all p values would be very small, so observing a p value as big as 0.05 would favour the null hypothesis. I’m not suggesting that the paper should be rewritten using likelihood ratios, or false positive risk, as the criterion for judging whether or not an observed difference is or isn’t a result of sampling error. But more than enough has been written on the limitations of the p value approach that it would be helpful to them reader to include a discussion of the Jeffreys-Lindley effect so that the assumptions made in the paper are clearer to readers.”*

These are absolutely critical points, and ones we were remiss in not clarifying. As the reviewer notes astutely, this is only truly a problem if one relies on pure significance testing. This of course is liable to mislead in many cases, but is still extremely common in biomedical science, often employed without cognisance of why it is being undertaken. We’ve rewritten part of the introduction section and discussion address the points the reviewer raises. In the introduction (abridged):

“Significance testing is perhaps the most widely used approach for hypothesis testing in biomedical science, and accordingly this is the focus of this work... Inferences drawn from naive significance testing, however, are fraught with pitfalls. Significance levels are arbitrary, and the misguided interpretation that $p < 0.05$ is a proxy for proof has been widely criticised [17]. Many experimenters still wrongly believe that the p -value is the probability that experimental results are due to chance, but this is not the case. Simply warning against this misinterpretation, however, has been deemed an

'abysmal failure' [18]. This is a problem likely compounded by the ease of modern statistics packages, which can readily run any test the user dictates, whether or not these are appropriate."

As the discussion on effect sizes has been altered substantially, we felt that table 1 was not required and potentially confusing, and this has now been removed from the text. Please see our response to point 11 for modifications in this vein to the discussion.

- 3. "Equation 3 Needs more explanation, and a reference. The quantities are not defined at the point eq 3 appears: μ is not defined until further down page 5, and σ is not defined until you get to the appendix. It might also be worth mentioning that effect sizes that are normalised by σ are not of any clinical relevance."*

Our apologies for this oversight, this was a consequence of reshuffling parts of the manuscript. This equation (now equation 2) now has all quantities defined.

- 4. "In figure 2, the value of ω is not clear at all. Shouldn't " $\mu - 2\omega\sigma$ " in the legend be written as $\mu - 2\sigma$ i.e. the value for $\omega = 2$?"*

The reviewer is absolutely right, our apologies for this outstanding Latex typo. The legend has now been clarified and simplified for ease of readability.

- 5. "page 6, line 42. And page 12, line 16. "Erf is the standard error function". The use of "standard error" here is potentially confusing. Wouldn't it be better to say something like "here erf() is the error function, defined as . . .". Also erf() is normally spelled with lowercase, like exp()."*

We are in full agreement with the reviewer on this point and have modified the text as such.

- 6. "page 6, line 51. " μ is the limit of the warped mean as sample population, n , tends to infinity.". Wouldn't "censored" be a better word than "warped"? And wouldn't it be better to describe it as the true value, insofar as it assumes that μ and σ are known. There is no need to describe it as a limit (in fact it would only be a limit if the estimator were unbiased). The estimate using sample estimates of μ and σ will be variable but it isn't obvious (to me) that it will be biased so I'm not sure that it's right to describe it as giving "minimum changes from the mean due to redaction".*

The reviewer is indeed corrected, and we have removed much of this in the new method section for clarity.

- 7. "6. page 7, line 38 "< -2 ω " -again ω is not defined (see para 3, above). The example would be much clearer if were put as follows. Before redaction the sample mean and standard error are *** and ***, hence $t_{df} = ***$ and $p = ***$. After redaction of the largest observation (which is the only observation which is more than two 2 sample standard deviations above the sample mean) these values become ***. Presenting it this way would make it much clearer the extent to which the increase in t after redaction results from changes in sample standard deviation and to what extent it results from changes in its numerator."*

We thank for reviewer for this suggestion, and we've now rewritten the results to demonstrate this clearly – please see modified text for full details.

8. *“Figure 4. Should the ordinate be reduction in tumour diameter, rather than tumour diameter? It’s not at all clear where the control groups appear in the examples. Likewise at the top of page 10, it’s not at all clear what “with 10 ± 6 mm tumour diameter” refers to.”*

This was entirely our issue for not being sufficiently clear in our wording – every example in this section is effectively an experimental set being compared against a known distribution. This is especially confusing with the murine example, and we’ve re-written for clarity:

“Results from this simulated experiment are contrasted with a known control distribution where mean tumour diameter is 10 ± 6 mm in untreated mice.”

9. Page 10, line 17. “theoretical $\mu = 6.53$ ” should presumably be μ . A bit more explanation of what this should be compared with would make this clearer. A bit of explanation of the gaps in the red curve in Figure 5 would help the reader a lot. I wonder how realistic this example is: who on earth would exclude patients “not surviving beyond 6 months”? What reason could be given for doing something so apparently silly as that?

These are excellent points – it would be extremely silly, but there are reported instances of it occurring and other related issues with selective trial management. To cover this, we have rewritten the section:

“Figure 5 shows the Kaplan-Meier survival curves (depicting the fraction of surviving patients) for the entirety of the sample, and for a situation when patients not surviving beyond 6 months are excluded from the analysis. This corresponds to $\omega = 1$. For the all-patient cohort, median survival is $\exp(\mu)$ days, or 17.1 months. When those surviving under 6 months are excluded, equation three yields $\mu = 6.53$, corresponding to a median survival time of 22.9 months. A distribution fit to the simulated scenario in figure 5 yielded a log-normal with $\mu = 6.64$, in close agreement with theoretical prediction. This is statistically significantly different from μ ($p < 0.001$) with effect size 0.29. Redaction of these patients would thus incorrectly lead an investigator to conclude that the intervention significantly increases survival time. It should be noted that such a redaction would be extremely poor practice, but inadvertent redactions could pivot on more subtle issues than survival time, such as exclusions due to a certain biomarker concentration or patient age.”

10. *“page 11, line 13. “Equation 4 in this work yields the ‘best-case’ in larger data sets, due to the inverse dependence of these metrics on sample size”. The meaning of this isn’t clear to me (see also para 5, above). Likewise “effect size is more often a better metric to assess impact of an intervention”. I can’t recall reading any paper in which the effect size is not stated, though sometimes it’s given only in standardised form which makes judgement of clinical significance difficult.”*

This line has since been retired, as it harkens back to the same issue the reviewer noted in point 6. Instead, we’ve simply clarified this in the results section to avoid confusion. Regarding effect size, please see point 12 below.

11. *“page 11, table 2. More verbal explanation of what the table tells you is needed. Line 31. “false significance is more readily found in larger data sets, due to the inverse dependence of these metrics on sample size”. That’s true only for the Fisherian approach, It isn’t true for either the likelihood ratio approach or the Bayesian approach (see para 1, above).”*

This is very true, and in line with the reviewer's comments in point 2, we've changed the discussion text on this point to read:

"Clearly, naive fixation on arbitrary statistical significance alone can dangerously mislead investigators, a fact that has been explicitly elucidated in recent years [17]. Significance testing in isolation can be misleading, largely because there is still widespread confusion over what significance actually means. Statistics arrived at must be seen in context; while large datasets are less prone to misleading bias than smaller ones, for example, false significance is more readily found in larger data sets, due to the inverse dependence of these metrics on sample size. Strictly speaking, this is only true for Fisherian approaches outlined in this work, and many of these pitfalls can be circumvented using different approaches to hypothesis testing, such as Bayesian analysis or Likelihood ratios [18]. Effect sizes too, should inform interpretation of results derived. For ascertaining clinical impacts, absolute effect sizes are much more important than arbitrary p-values. While these are by no means new observations, the low levels of replicable research in biomedical science suggests lessons still must be learnt."

12. *"Line 34. "effect size is more often a better metric to assess impact of an intervention". Here (and elsewhere) it should be made clear whether "effect size" refers to normalised effect size or absolute effect size. Only the latter is useful for judging clinical significance. The former is much like saying that the z score is a "better metric to assess impact of an intervention". I'm not sure how useful this comment is."*

Please see reply to point 11 above.

13. *"page 12, line 4. "redacting data points from one group, and appending them to other ...". A reference to the fragility index might be appropriate here."*

We thank the reviewer for making us aware of this measure! We have added the following:

"Analysis using the fragility index has shown this to be an issue in the interpretation of a335number of clinical trial results [40, 41]."

14. I've tried to evaluate eqs 6 and 7 in R. I'd like to check against the authors' code, but it doesn't seem to be available.

We are happy to upload any code the reviewer thinks would be useful. The authors do not use R very often, but they have uploaded a version of this in Mathematica for the reviewer, in notebook format. The online Mathematica player is here and the code is uploaded with this revision. A step guide is shown below for convenience:

pdf of normal distribution :

$$\text{In[39]:= } g = \frac{1}{\sigma \sqrt{2\pi}} \text{Exp}\left[\frac{-(x-\mu)^2}{2(\sigma^2)}\right];$$

Constant term for scaling (Equation 5)

```
In[40]:= cdenom = Assuming[σ > 0, Integrate[g, {x, (μ - w*σ), Infinity}]];
cnom = Assuming[σ > 0, Integrate[g, {x, -Infinity, Infinity}]];
cterm = Simplify[cnom / cdenom]
```

$$\text{Out[42]= } \frac{2}{1 + \text{Erf}\left[\frac{w}{\sqrt{2}}\right]}$$

Unscaled mean expression

```
In[44]:= expectation = Assuming[σ > 0, Integrate[g*x, {x, (μ - w*σ), Infinity}]]
```

$$\text{Out[44]= } \frac{1}{2} \left(\mu + e^{-\frac{w^2}{2}} \sqrt{\frac{2}{\pi}} \sigma + \mu \text{Erf}\left[\frac{w}{\sqrt{2}}\right] \right)$$

+

Final expression for modified mean :

```
In[43]:= μn = FullSimplify[expectation * cterm]
```

$$\text{Out[43]= } \mu + \frac{e^{-\frac{w^2}{2}} \sqrt{\frac{2}{\pi}} \sigma}{1 + \text{Erf}\left[\frac{w}{\sqrt{2}}\right]}$$

15. *“page 14, line 45. Clearly eq. 9 is a measure of skewness (though not the conventional measure), but please explain why it is “a measure of how many patients are required in total for a trial to generate a spurious result”*

This line is very misleading and superfluous – it has now been removed!

16. *“page 15. I’ll admit to having to struggle a bit to follow the results of the simulations. This is partly a result of the curious properties of ω as a measure of censorship -no censoring corresponds to ω → ∞. More importantly, it’s partly a result of the fact that it isn’t explained clearly exactly what is tested in the t tests. Is the sample from the uncensored distribution tested against the sample from the censored population in a two-sample t test?”*

We were reluctant to use the word censoring giving the clinician application of this word in trials, but the reviewer is absolutely correct in their interpretation. We have added text to reflect this no.w

17. *“Figure 6 is presumably calculated, not found via simulation, so it should not appear under the simulation heading. Please move it earlier, together with the first paragraph on page 15, and*

provide a verbal description of what it shows. Also please say exactly how the curves are calculated. It's also worth mentioning again that the results hold only if you are happy to ignore the Jeffreys-Lindley effect and the likelihood ratio approach."

This has now been clarified in the appendix text.

18. "Figure 7. Again please provide a longer verbal description of what the Figure shows. I wonder whether it's sensible to start the curves at $\omega = 0$. It's hard to imagine anyone redacting all the data below the mean."

This is very true, but we have left it in for completeness, noting it is :

"..likely unrealistic but given for completeness."

Minor points also considered and corrected

Appendix C

Reply to reviewers and editors

Davidrobert.grimes@dcu.ie

28th October 2021

Dear Reviewers and editors,

Thank you ever so much for your continued insight and patience. We are grateful for the effort you've put into this work, and we hope our revisions reflect this. For brevity, I have listed all relevant responses and changes below. Thank you again.

Yours, on behalf of the authors,

David Robert Grimes

Responses to reviewer 1

1. *"The revision dealt well with many issues raised by myself and the other reviewer, but the current version remains insufficient in how it refers to and discusses the wider literature on truncation models, outlier removal, and robust statistics. For instance, it fails to discuss Barnett & Lewis (1978, 1994) which was cited over 7000 times in its different editions (in Google Scholar and many in the biomedical literature) or similar work that has been deeply influential but not referred to in the paper. It also ignores the work on robust statistics (e.g., Huber, 2004; cited over 20000 times!) and other important and influential literatures on how to deal properly with non-normality in data. Much of the current problems have been addressed dozens of times, and this needs to be acknowledged and discussed. Surely, it does not hurt to reinstate the important message that biases emerge from the very practices that the authors address but it is unacceptable to pretend the current readers all are sitting on a tiny island of the literature where they somehow missed the relevant literatures that thousands of others have cited and multiple dozens of other have actually already investigated to great lengths."*

We thank the reviewer again for their insight – it is true that too often in research, we end up re-inventing the wheel, especially in interdisciplinary science. In light of this, we have taken the reviewer's suggestion on board, and pointed out explicitly that such problems have been considered in terms of robust statistics and outlier analysis, and we have now updated the text to reflect this.

Responses to reviewer 2

1. *"The use of the term 'atrophied' seems inaccurate to me. I suggest that it should be replaced."*

We agree with the reviewer this might be misleading – we have subsequently rephrased this section, using the term "removed" instead.

2. *"The revision at lines 300-312 is a great improvement, I think. But surely it is not sensible to say "strictly speaking". It could be made even more clear if it were pointed out that the difference between the effect of sample size is dependent on whether you calculate the likelihood ratio by the p-equals method or the p-less-than method. Bayesian use the former*

but the distinction doesn't depend on whether you are Bayesian or not. It depends on whether you think that the analysis should reflect the evidence provided by the experiment you're analysing (p-equals) or long-term behaviour (p-less-than)."

We thank the reviewer for their clarity and insight – we agree, the strictly speaking clause is not required. We have accordingly removed it to reflect this.

We thank both reviewers and the editorial team again for their insight.